# Filters, Thresholds, and Geodesic Distances for Scribble-based Interactive Segmentation of Medical Images

*Zdravko Marinov[1,2] ⓘ, Alexander Jaus[1,2] ⓘ, Jens Kleesiek[3,4] ⓘ, and Rainer Stiefelhagen[1] ⓘ

[1] Karlsruhe Institute of Technology, Karlsruhe, Germany
[2] HIDSS4Health - Helmholtz Information and Data Science School for Health, Karlsruhe/Heidelberg, Germany
[3] Institute for AI in Medicine, University Hospital Essen, Essen, Germany
[4] Cancer Research Center Cologne Essen (CCCE), University Medicine Essen, Essen,
*Corresponding Author: `zdravko.marinov@kit.edu`

**Abstract.** Interactive segmentation plays a vital role in medical image analysis, facilitating accurate diagnosis and treatment planning through real-time interaction and rapid annotations. Scribble-based methods, where users draw over target structures, are particularly effective for delineating thin structures like vessels, providing precise pixel-level detail compared to bounding boxes. MedSAM, introduced in 2023, is optimized for bounding box inputs, which limits its effectiveness for precise interaction types such as scribbles. Additionally, it exhibits a slower inference due to its large size. To address these limitations, we evaluated simpler models such as thresholding, Meijering filters, and Geodesic Distance Transforms. These models outperformed MedSAM in segmentation accuracy and efficiency across fundus, microscopy, PET, and OCT, achieving a Dice Score of 62.31 and a Normalized Surface Dice of 67.01 on the validation set. Our findings highlight the effectiveness of traditional methods and reveal the current limitations of emerging foundation models. This comparative analysis aims to improve MedSAM's robustness and efficiency, contributing to the development of a more reliable general model for medical image segmentation.

## 1 Introduction

**Background.** Deep learning advancements have significantly enhanced the segmentation of anatomical structures and lesions in medical images. These models, however, often depend on manually annotated datasets, which are labor-intensive to create [10,32,2,21,41,17]. To reduce this burden, interactive segmentation methods have been developed, utilizing simpler annotations like clicks or bounding boxes instead of detailed voxelwise labels [29,44,14,15,3,40,25,39,30]. These methods combine user inputs with image data to make predictions, which, once verified by medical professionals, can be used as new annotations [29]. However, scribbles provide a more intuitive and versatile annotation method, allowing

users to specify exact pixels of the target object. Various studies have employed freehand strokes to initiate segmentation masks [13,6,42,40] and to refine existing ones [13,42], significantly speeding up the annotation process. Interactive models leveraging scribbles can therefore facilitate quicker and more precise data annotation.

**Related Work.** MedSAM [26] is a fine-tuned version of the Segment Anything Model (SAM) [24] that has been trained on 11 imaging modalities and over 1.5 million image-mask pairs, showcasing impressive generalizability across various segmentation tasks [26,29]. However, MedSAM is limited to processing bounding boxes or clicks and does not support scribble-based inputs. Adaptations like ScribblePrompt [42] have optimized MedSAM for scribble guidance, maintaining its generalization across multiple modalities. Despite these advancements, the research on scribble-based interactions remains relatively sparse. To address this gap, Ma et al. [26] organized the Segment Anything in Medical Images on Laptop Challenge (Task 2: Scribble) [5] to develop and evaluate efficient scribble-based methods for interactive segmentation. This paper presents our submission to this challenge, aiming to contribute further to the exploration of scribble-based interactive segmentation techniques.

**Motivation.** Our motivation stems from the efficacy of classical methods in addressing segmentation tasks with simplicity and efficiency. By comparing these methods against recent generalist models like MedSAM, we seek to unravel insights into how such straightforward models can surpass large, pre-trained vision models. Furthermore, we aim to initiate discourse on enhancing MedSAM for future iterations. Our study offers the following contributions:

1. We investigate classical approaches for the fundus, microscopy, PET, and OCT modalities and investigate if they can outperform MedSAM's lightweight implementation (LiteMedSAM-Scribble[6]) in terms of segmentation accuracy and efficiency
2. We examine the failure cases and discuss why MedSAM silently fails on certain modalities and propose how to tackle this in future fine-tuning iterations
3. We make all our code and trained models publicly available to the community

## 2    Method

We go over the fundus, OCT, PET, and microscopy imaging modalities one-by-one and examine which classical approaches are able to outperform MedSAM and propose techniques to make MedSAM more efficient on modalities on which we could not outperform it. For the rest of the modalities in the challenge, we use MedSAM's lightweight implementation LiteMedSAM-Scribble.

**Note:** We only focus on the segmentation tasks seen in the MedSAM training dataset, e.g., only FDG-PET lesions segmentation and only vessel segmentation on fundus images. We also always use LiteMedSAM-Scribble as a lightweight MedSAM implementation and refer to it as MedSAM for brevity.

---

[5] https://www.codabench.org/competitions/2566/

[6] https://github.com/bowang-lab/MedSAM/tree/LiteMedSAMScribble

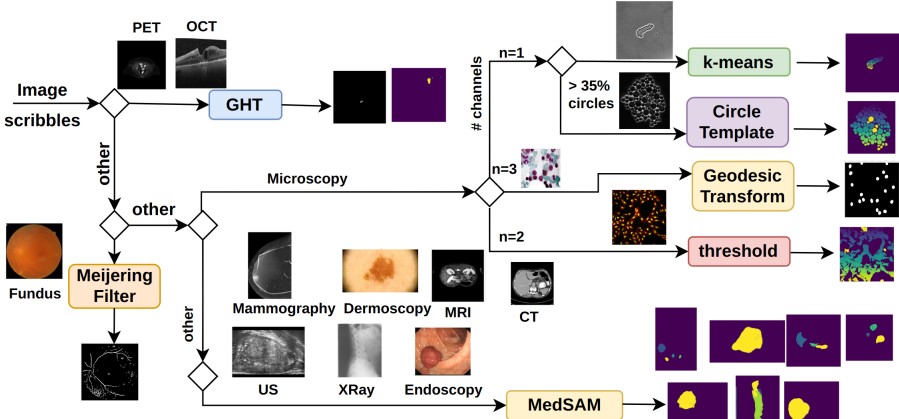

Fig. 1: Overview of our pipeline. We apply a different model for the various imaging modalities. For PET and OCT, we use the generalized histogram threshold (GHT) [4]. For microscopy, we check the number of channels and apply: (1) Geodesic Distance Transform [7] for $n = 3$; (2) a threshold, set to the mean image intensity for $n = 2$; (3) k-means clustering [28] for $n = 1$ if less than 35% of the bounding boxes contain circular objects, otherwise we apply a circle template matching. For fundus images, we apply a Meijering filter [31]. For the rest of the modalities, we use MedSAM.

## 2.1  PET and OCT

**Tasks:** The challenge presents PET data exclusively sourced from the AutoPET dataset [10], which is dedicated to the delineation of active tumor lesions across the whole body using Fludeoxyglucose (FDG) as the imaging agent. Similarly, the OCT data is drawn solely from a single dataset [1], concentrating on the segmentation of intraretinal cystoid fluid.

**Challenges:** The scarcity of public PET datasets for tumor segmentation [10,35,12] hampers the development of large-scale foundational models in this field. In the AutoPET dataset [11], PET lesions exhibit low contrast against surrounding tissues, and other healthy anatomical structures, such as the heart, brain, and bladder, also show high physiological uptake. Moreover, the top results from AutoPET 2023[7] are modest, with the highest Dice Score being 0.36, highlighting the difficulty of the task. For OCT, the images have high resolution but feature tiny target structures, resulting in a pronounced class imbalance.

**Classical Approaches:** Thresholding techniques are commonly employed for tumor segmentation in PET scans [22,20,33], delivering promising results due to their simplicity and intuitive application. Scribble-based methods enhance this approach by allowing thresholds to be applied within the local context of the scribble, effectively excluding healthy tissues like the heart and brain that lie

---

[7] https://autopet-ii.grand-challenge.org/leaderboard/

outside the defined boundaries. This is in contrast to previous methods that globally excluded regions such as the brain and bladder from the whole-body context [34,10,16]. Furthermore, using scribbles in OCT images helps mitigate class imbalance by focusing on the local area around the scribble, rather than the entire high-resolution image.

To effectively utilize the context around the scribbles for thresholding, we first convert the scribbles into bounding boxes by expanding them by 50 pixels in all directions. The bounding box $B$ is computed as:

$$B = \{\min(S_x) - 50, \min(S_y) - 50, \max(S_x) + 50, \max(S_y) + 50\} \qquad (1)$$

where $S_x, S_y$ are the sets of $x$ and $y$ coordinates of all the provided scribbles.

Next, we calculate the Generalized Histogram Threshold (GHT) [4] by aggregating PET or OCT values from all expanded bounding boxes. This threshold is then applied to the entire volume to generate a unified prediction, with instance indices matching their respective scribble indices. For PET images, we retain values exceeding the threshold, as tumors exhibit high FDG uptake. In contrast, for OCT images, we keep values below the threshold due to the darker appearance of cystoid fluids.

### 2.2   Microscopy

**Tasks:** MedSAM's microscopy training data is exclusively derived from the NeurIPS 2022 CellSeg dataset [27]. This dataset poses significant challenges due to its diversity, containing images captured with various microscope types such as brightfield, fluorescent, phase-contrast (PC), and differential interference contrast (DIC). Additionally, the dataset includes a variety of cell types as segmentation targets, further increasing the complexity of the task.

**Challenges:** The microscopy imaging modality presents several significant challenges: (1) The number of instances per image can be exceptionally high, sometimes exceeding 1000, leading to substantial computational overhead when processing each bounding box individually. (2) The diversity of microscope types requires either a highly robust generalist model or multiple specialized models to handle the varying imaging characteristics effectively. (3) The high-resolution nature of the images is problematic, as critical details may be lost when resizing to smaller resolutions, such as MedSAM's resizing to $256 \times 256$.

**Classical Approaches:** We employ different classical methods based on the number of channels in the image. Additionally, we convert all scribbles into bounding boxes by expanding the scribble coordinates by 50 pixels in all directions, as described in Eq. 1.

**Grayscale:** When processing grayscale images, we apply a k-means clustering algorithm [28] with $k = 2$. To identify the foreground class, we analyze the pixel frequency for each class within a $10 \times 10$ window at the center of the bounding box, selecting the class with the higher pixel count. We use the Hough circle transform [23] to detect circles of various radii within each bounding box and for images where $> 35\%$ of the bounding boxes contain circular objects the largest detected circle serves as the prediction for that bounding box.

**Two-channel:** For images with two channels (commonly three channels in practice, where one channel is filled with zeros, as seen in fluorescent microscopy), we employ the mean image intensity as a threshold. This simple yet effective method enables reliable segmentation.

**RGB:** For RGB images, we opt to utilize the scribbles as seeds for a Geodesic Distance Transform (GDT) [7]. For each scribble, we first transform it into a bounding box $B$ and then apply the GDT within the bounding-box cropped image $I_B$ with the scribble $S$ used as seeds:

$$\text{GDT}(x, S, I_B)_{x \in I_B} = \min_{x' \in S} d(x, x')$$

(2)

where $d(x, x')$ is the Geodesic distance as described in [7]. We utilize the GeodisTK[8] implementation for computing the Geodesic distance. The Geodesic distance quantifies the minimum cost of traversing from one pixel to another, considering the intensity gradients along the path, which aids in edge-aware segmentation. To apply it for segmentation, we assign all pixels $x$ above the 70th percentile $\text{GDT}_{70}$ as background and the rest as foreground.

$$\text{seg}(x) = \begin{cases} 0 & \text{if } \text{GDT}(x, S, I_B) > \text{GDT}_{70}, \\ 1 & \text{otherwise.} \end{cases}$$

(3)

### 2.3   Fundus

**Tasks:** Fundus tasks concentrate on optic discs and cups [37] as well as more intricate structures such as retinal vessels [38,8]. In the case of vessels, bounding-box interaction signals prove inadequate for highlighting relevant context, however, scribbles are precise enough to indicate the underlying tubular segmentation targets.

**Challenges:** Fundus image datasets focusing on retinal vessels exhibit a large diversity in terms of labeling protocols and imaging characteristics, leading to a large domain shift between datasets [9]. In particular, MedSAM struggles with thin structures such as vessels as it has only been trained on optic disc and cups in the fundus domain and bounding boxes are a suboptimal guidance signal for such structures.

**Classical Approaches:** To detect tubular-like structures, we apply a Meijering filter [31] over the whole image $I$ and assign all pixel values $x$ between the 90th and 95th percentile to the "foreground" as:

$$\text{seg}(x) = \begin{cases} 1 & \text{if } \text{Meijering}(I)_{90} < x < \text{Meijering}(I)_{95}, \\ 0 & \text{otherwise.} \end{cases}$$

We chose this interval as values above the 90th percentile have a high enough filter response to be considered tubular structures, but we observed that values above the 95th percentile correspond to noise.

---

[8] https://github.com/taigw/GeodisTK

## 2.4   Other modalities

For the rest of the modalities (CT, MRI, X-Ray, Mammography, Endoscopy, Dermoscopy, Ultrasound), due to time constraints, we simply utilized MedSAM's lightweight pre-trained implementation LiteMedSAM-Scribble[9]. Note, that for this challenge, only 2D images from the CT and MRI domain were used.

Table 1: Summary of our used models for the final submission.

| Modality | Used Model |
|---|---|
| PET and OCT | Generalized Histogram Threshold [4] |
| Fundus | Meijering Filter [31] |
| Microscopy | k-means, Thresholding, Geodesic Distance [7], Circle Templates [23] |
| Other | LiteMedSAM [26] |

## 2.5   Preprocessing

We re-used the code provided by LiteMedSAM-Scribble[10] for loading the data and inferring predictions and added more functions to the script for our methods. We avoid loading LiteMedSAM's weights for tasks which do not need it and import modules only immediately before they are used. The image loading and preprocessing is done as follows:

**LiteMedSAM:** The image is resized to a common size of $256 \times 256$ and padded to the shorter side to keep the original aspect ratio. Then, the image is min-max normalized and fed to the model. The model performs a forward pass for each scribble.

**k-means and Geodesic Distance Transform (GDT):** When applying k-means clustering or GDT, we use the unnormalized values within each transformed bounding box (computing via Eq. 1 for each scribble) and apply k-means/GDT for each instance.

**Thresholding:** Thresholds are always computed using the combination of all image values in the bounding boxes and then applied to the whole unnormalized image. Instance indices are then assigned according to the scribble indices.

**Meijering filter:** The Meijering filter is applied on the whole unnormalized image. However, the image is first converted to grayscale by computing the mean over all of its channels.

---

[9] https://github.com/bowang-lab/MedSAM/tree/LiteMedSAMScribble
[10] https://github.com/bowang-lab/MedSAM/blob/LiteMedSAMScribble/CVPR24_
LiteMedSAM_infer_scribble.py

### 2.6   Post-processing

After each forward pass, we perform only one post-processing transformation. For all images, we keep the largest connected component and fill all the holes within it.

## 3   Experiments

### 3.1   Dataset and evaluation measures

We used only the challenge dataset for model development and validation. The evaluation metrics include two accuracy measures—Dice Similarity Coefficient (DSC) and Normalized Surface Dice (NSD), alongside one efficiency measure: running time. These metrics collectively contribute to the ranking computation.

### 3.2   Implementation details

**Environment settings**  The development environments and requirements for all our methods are presented in Table 2.

Table 2: Development environments and requirements for all our methods.

| | |
|---|---|
| System | Ubuntu 22.04.4 LTS |
| CPU[11] | Intel(R) Core(TM) i7-13700H CPU@5.00GHz |
| RAM | 8×4GB; 5200MT/s |
| GPU (number and type) | None |
| CUDA version | 11.8 |
| Programming language | Python 3.10.14 |
| Deep learning framework | torch 2.2.1 |
| Specific dependencies | None |
| Code | `https://github.com/Zrrr1997/medsam_cvhci_scribble` |

**Training protocols**  For MedSAM, we use the provided pre-trained LiteMedSAM-Scribble model whose training is described in [26]. As such, we did no training or fine-tuning whatsoever since we implemented only classical non-deep learning methods for our submission.

## 4   Results and discussion

We discuss the results of the individual modalities one-by-one as we propose different models for the OCT, PET, fundus, and micrsocopy imaging modalities.

---

[11] `https://ark.intel.com/content/www/us/en/ark/products/232128/intel-core-i7-13700h-processor-24m-cache-up-to-5-00-ghz.html`

Table 3: Training protocol for LiteMedSAM.

| Pre-trained Model | LiteMedSAM [26] |
|---|---|
| Batch size | No further fine-tuning |
| Patch size | $256{\times}256{\times}3$ |
| Total epochs | No further fine-tuning |
| Optimizer | No further fine-tuning |
| Initial learning rate (lr) | No further fine-tuning |
| Lr decay schedule | No further fine-tuning |
| Training time | No further fine-tuning |
| Loss function | No further fine-tuning |
| Number of model parameters | 9.79M[12] |
| Number of flops | 1.81G[13] |
| $CO_2$eq | No further fine-tuning |

## 4.1   PET and OCT

**Efficiency Strategies:** Thresholding eliminates the need for slow forward passes as the threshold is applied directly on the whole image in a single binary operation. Table 4 shows that the thresholding is extremely fast for both the PET and OCT modalities although it sacrifices some of the performance on the PET modality. We do not report the Dice and NSD for OCT as it is not part of the validation set but we report its efficiency on the training data. For this, we simply simulate empty OCT scribbles to measure the inference time.

Table 4: Results on the validation stage of the challenge for PET. We cannot report results for OCT as there are no validation images for that domain but we do report the time per image on the training set [1].

| Model | Dice | NSD | Time per Image |
|---|---|---|---|
| LiteMedSAM (PET) | 66.80 | 49.42 | 0.01s |
| Thresholding (PET) | 49.00 | 67.00 | 0.77s |
| LiteMedSAM (OCT) | - | - | 1.78s |
| Thresholding (OCT) | - | - | 0.005s |

**How to improve MedSAM on PET and OCT?** The findings are alarming, as simple thresholding outperforms MedSAM in the PET task. We hypothesize that this discrepancy stems from the insufficiency of available data to enhance MedSAM's capacity for generalization and feature extraction sufficiently.

---

[12] https://github.com/sksq96/pytorch-summary
[13] https://github.com/facebookresearch/fvcore

Moreover, PET data could benefit from augmentation with anatomical labels derived from paired CT scans [19,34], which was the winning approach in AutoPET 2023. This approach supplements the model with expert knowledge regarding affected anatomical regions, thereby enriching its understanding. Similarly, in the OCT domain, incorporating additional anatomical labels, such as those corresponding to various retinal layers, could lead to substantial improvements.

### 4.2   Microscopy

**Efficiency Strategies:** The classical approaches that we apply improve MedSAM's efficiency 15-fold. The segmentation results on the validation set are also x3 higher as seen in Table 5.

Table 5: Results on the validation stage of the challenge for Microscopy images

| Model | Dice | NSD | Time per Image |
|---|---|---|---|
| LiteMedSAM | 12.00 | 9.00 | 38.2s |
| k-means OR threshold OR LiteMedSAM | 31.81 | 32.55 | 2.4s |

**How to improve MedSAM on Microscopy?** It appears that MedSAM encounters challenges with miniature scribbles, particularly when they are resized to fit its input size of $256 \times 256$. We believe that adopting a crop-then-infer approach could enhance MedSAM's performance. By cropping the image before inference, only the crop would require resizing, ensuring that the scribbles retain their detail and enabling MedSAM to focus more effectively on the local instance. However, further research is warranted to validate this hypothesis thoroughly.

### 4.3   Fundus

**Efficiency Strategies:** We focus on segmentation accuracy rather than efficiency in this domain. Thus, our model based on the Meijering filter [31] is slower than the baseline (LiteMedSAM-Scribble), but achieves a much higher Dice score and NSD as seen in Tab. 6.

| Model | Dice | NSD | Time per Image |
|---|---|---|---|
| LiteMedSAM | 5.00 | 0.00 | 1.2s |
| Meijering filter | 38.00 | 49.00 | 9.7s |

Table 6: Results on the validation stage of the challenge for Fundus images

**How to improve MedSAM on Fundus?** MedSAM is only trained on the tasks of optic disc and optic cup segmentation for fundus images [26]. As such,

there is a domain/task gap when it is asked to segment thin vessel-like structures. However, there are many public fundus datasets with annotated retinal vessels [8,38,9,5,18,36] that could be used to fine-tune MedSAM on this specific task in future iterations.

### 4.4   Quantitative results on validation set

Table 7 shows that our approach outperforms (on average) the baseline (LiteMed-SAM). However, we also show in Tables 4-5 that we are able to significantly improve the efficiency of the baseline on OCT, PET, and microscopy images.

Table 7: Quantitative evaluation results. Baseline: LiteMedSAM-Scribble. Ablations were done on the Geodesic percentile threshold in Equation 3.

| Target | Baseline | | Ablation $GDT_{10}$ | | Ablation $GDT_{90}$ | | Proposed $GDT_{70}$ | |
|---|---|---|---|---|---|---|---|---|
| | DSC(%) | NSD(%) | DSC(%) | NSD(%) | DSC(%) | NSD (%) | DSC(%) | NSD (%) |
| CT | 81.00 | 83.00 | 81.00 | 83.00 | 81.00 | 83.00 | 81.00 | 83.00 |
| MR | 70.00 | 77.00 | 70.00 | 77.00 | 70.00 | 77.00 | 70.00 | 77.00 |
| PET | 67.00 | 90.00 | 49.00 | 67.00 | 49.00 | 67.00 | 49.00 | 67.00 |
| US | 85.00 | 88.00 | 85.00 | 88.00 | 85.00 | 88.00 | 85.00 | 88.00 |
| X-Ray | 22.00 | 19.00 | 22.00 | 19.00 | 22.00 | 19.00 | 22.00 | 19.00 |
| Dermoscopy | 90.00 | 91.00 | 90.00 | 91.00 | 90.00 | 91.00 | 90.00 | 91.00 |
| Endoscopy | 94.00 | 97.00 | 94.00 | 97.00 | 94.00 | 97.00 | 94.00 | 97.00 |
| Fundus | 5.00 | 0.00 | 38.00 | 49.00 | 38.00 | 49.00 | 38.00 | 49.00 |
| Microscopy | 12.00 | 9.00 | 22.44 | 24.12 | 30.22 | 30.78 | 31.81 | 32.55 |
| Average | 58.00 | 62.00 | 61.27 | 66.12 | 62.14 | 66.86 | 62.31 | 67.01 |

### 4.5   Segmentation efficiency results on validation set

The efficiency on a few samples from the validation set are listed in Table 8. Our optimization on the PET and microscopy modalities contributes to a much more efficient prediction time, especially when there are many instances in the image such as for 2D_Microscope_0016 where the inference is reduces from 127 seconds to only 6 seconds with our approach. Our ablation regarding the Geodesic percentile also demonstrates negligible differences in terms of efficiency.

### 4.6   Qualitative results on validation set

We show some qualitative image examples for the predictions of our models on various modalities.

**PET and OCT threshold-based methods.** Fig. 2 shows examples of predictions on OCT and PET images. Although thresholds are more efficient than MedSAM, they do produce noisier predictions.

Table 8: Quantitative evaluation of segmentation efficiency in terms of running time (s) on the hardware specified in Table 2. Abl.: Ablation

| Case ID | Size | #Objects | Baseline | Abl. $GDT_{90}$ | Ours $GDT_{70}$ |
|---|---|---|---|---|---|
| 2D_US_0525 | (256, 256, 3) | 1 | 1.9 | 1.9 | 1.9 |
| 2D_X-Ray_0001 | (2487, 2048, 3) | 7 | 6.0 | 6.0 | 6.0 |
| 2D_Dermoscopy_0003 | (3024, 4032, 3) | 1 | 3.7 | 3.7 | 3.7 |
| 2D_Endoscopy_0086 | (480, 560, 3) | 1 | 2.1 | 2.1 | 2.1 |
| 2D_Fundus_0003 | (2048, 2048, 3) | 1 | 2.5 | 9.9 | 9.9 |
| 2D_Microscope_0008 | (1536, 2040, 3) | 19 | 2.6 | 1.9 | 1.8 |
| 2D_Microscope_0016 | (1920, 2560, 3) | 241 | 126.8 | 6.3 | 5.7 |
| 2D_PET_0001 | (200, 200, 3) | 1 | 1.9 | 0.06 | 0.06 |
| 2D_PET_0002 | (192, 192, 3) | 1 | 1.9 | 0.08 | 0.08 |

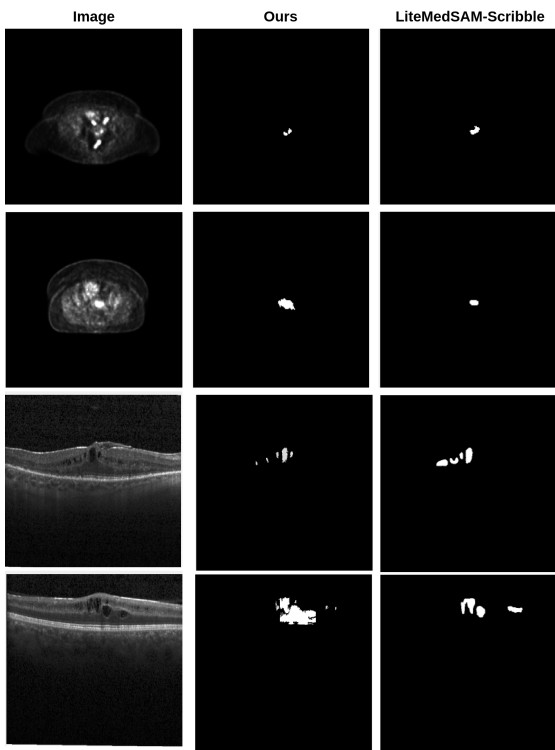

Fig. 2: Examples for OCT and PET predictions.

**Microscopy classical approaches.** Fig. 3 demonstrate failure cases of MedSAM in the microscopy domain. It seems that MedSAM struggles with small structures with ambiguous boundaries. MedSAM also struggles with multiple instances indicated with scribbles and focuses only on a small subset of them, or even only on one (last two rows). In contrast, classical approaches per-

form quite well in these domains, landing a spot in our methodology for our final submission to the challenge.

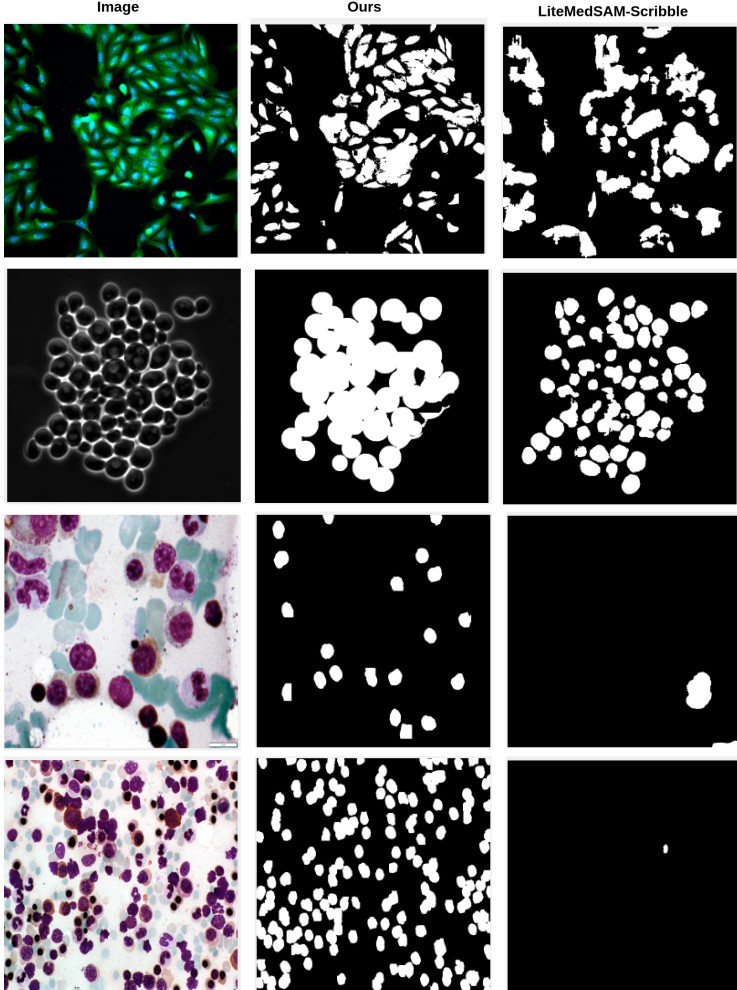

Fig. 3: Examples of microscopy predictions from the validation set.

**Fundus.** Fig. 4 shows examples for predictions on fundus images. Although the filter produces noisy artifacts, it does segment a large portion of the retinal vessels. MedSAM, on the other hand, clearly focuses on the optic disc and cup as the model has no notion of what the target task is since it has only been trained on segmenting discs and cups. This bias can be alleviated by either fine-tuning MedSAM on more datasets, containing new tasks, such as vessel segmentation,

or to provide contextual information on what is supposed to be segmented in the image, e.g., with an additional "task prompt".

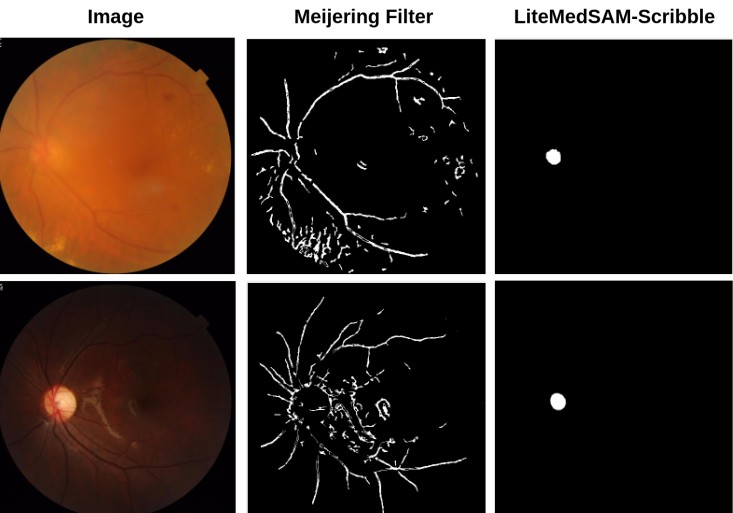

Fig. 4: Examples for Fundus predictions

### 4.7   Results on final testing set

The results on the final test set are presented in Table 9. Our focus on classical methods is evident in the leading runtimes on the leaderboard, where we secured 1st place across all modalities. By making only minor adjustments to the post-processing pipeline of LiteMedSAMScribble, our submission significantly outperforms the baseline in the CT and X-Ray domains. However, this improvement is not observed in the MR, Endoscopy, and Ultrasound modalities. We ranked last in the Fundus domain, possibly due to our assumption that all images contain vessel-like structures as targets, thereby overlooking other features, such as optic discs. Overall, our method achieved 2nd place on the final leaderboard out of a total of four methods.

s

### 4.8   Post-challenge Analysis

During the post-challenge phase, we have participated in the **Performance Booster** track without the use of any external datasets. Here, we describe how we have improved our methods.

Table 9: Results on the testing stage of the challenge. Worst rank is 4

| Modality | Dice (Rank) | NSD (Rank) | Runtime (Rank) |
|---|---|---|---|
| CT | 68.86 (1) | 74.87 (1) | 5.37 (1) |
| MR | 49.60 (3) | 53.88 (3) | 5.08 (1) |
| X-Ray | 56.07 (1) | 64.48 (2) | 5.80 (1) |
| Endoscopy | 83.69 (3) | 85.78 (3) | 5.09 (1) |
| Fundus | 0.00 (4) | 0.00 (4) | 15.48 (1) |
| Microscopy | 37.03 (2) | 38.29 (2) | 1.36 (1) |
| OCT | 22.11 (1) | 23.12 (3) | 1.08 (1) |
| PET | 65.17 (3) | 77.80 (3) | 0.97 (1) |
| US | 60.89 (3) | 61.36 (3) | 5.26 (1) |

**Changed Methodology** We made only a few adjustments to our methods, focusing on incremental improvements. For MR, Endoscopy, and Ultrasound, we remove all of our previous post-processing as we observed as decline in performance in Table 7. Fo the Fundus domain, we distinguish between vessels and optic discs as a segmentation target by counting the number of scribbles. If the count is below 4, we simply use the baseline (LiteMedSAMScribble), otherwise, we use our Meijering filtering approach.

**Results from Post-challenge Analysis** The changes in the code led to an improvement in the runtime for the MR, Endoscopy, and Ultrasound domains in terms of runtime as we omitted the postprocessing. For Fundus, we see a dramatic improvement of 79% Dice and 81% NSD. The changes in performance are listed in Table 10.

Table 10: Post-challenge changes in performance

| Modality | Dice | NSD | Runtime |
|---|---|---|---|
| US | 60.89 | 61.36 | 5.04 |
| MR | 49.60 | 53.88 | 5.00 |
| Endoscopy | 83.69 | 85.78 | 4.94 |
| Fundus | 79.46 | 81.43 | 6.05 |

### 4.9   Limitation and future work

Our methodology has two primary limitations: (1) it concentrates on specialized models tailored to individual modalities rather than developing a model capable of reasoning and segmenting structures across any modality, thus restricting our approach's generalizability; (2) our models rely on strong assumptions about the underlying task, such as vessel segmentation in fundus images, and if the task

on the test set changes, our models are likely to fail completely, as observed with MedSAM in Fig. **??**. However, we deliberately use this segmented approach to identify and understand the weaknesses in MedSAM, with the goal of improving it in future iterations. Our findings indicate that incorporating explicit assumptions about imaging modalities can provide a robust signal, sometimes outperforming MedSAM in specific cases, but it needs to be applied adaptively.

A promising future direction involves adding an adaptive task prompt to MedSAM. This would act as a model or an additional input that provides information about the underlying task, such as "fundus vessel segmentation" or "fundus optic cup segmentation." This can potentially adjust MedSAM's mode to the specific task, producing more accurate predictions and avoiding the training biases seen in Fig. **??**. In essence, informing the model about the specific segmentation task should enhance its adaptability and performance. This approach allows the integration of domain knowledge directly into the model, potentially tailoring it to specific domains and tasks.

## 5    Conclusion

Our results suggest that MedSAM needs further fine-tuning on scribble-based data and a method for adapting to new tasks it has not encountered, such as vessel segmentation. Our findings reveal the importance of integrating explicit task knowledge to surpass MedSAM's current performance. We propose for future iterations that a task adapter, which supplies information about the target structure and imaging modality, could boost MedSAM's effectiveness in these challenging areas.

**Acknowledgements** We thank all the data owners for making the medical images publicly available and CodaLab [43] for hosting the challenge platform. The present contribution is supported by the Helmholtz Association under the joint research school "HIDSS4Health – Helmholtz Information and Data Science School for Health. Parts of this work were performed on the HoreKa supercomputer funded by the Ministry of Science, Research and the Arts Baden-Württemberg and by the Federal Ministry of Education and Research.

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

Table 11: Checklist Table. Please fill out this checklist table in the answer column.

| Requirements | Answer |
| --- | --- |
| A meaningful title | Yes |
| The number of authors ($\leq 6$) | 4 |
| Author affiliations and ORCID | Yes |
| Corresponding author email is presented | Yes |
| Validation scores are presented in the abstract | Yes |
| Introduction includes at least three parts: background, related work, and motivation | Yes |
| A pipeline/network figure is provided | Figure 1 |
| Pre-processing | Page 8 |
| Strategies to data augmentation | Page 10 |
| Strategies to improve model inference | Pages 3-8 |
| Post-processing | Page 9 |
| Environment setting table is provided | Tables 2 and 3 |
| Training protocol table is provided | Table 3 |
| Ablation study | Page 11, Table 7 |
| Efficiency evaluation results are provided | Table 8 |
| Visualized segmentation example is provided | Figures 2-4 |
| Limitation and future work are presented | Yes |
| Reference format is consistent. | Yes |
| Main text $>= 8$ pages (not include references and appendix) | Yes |