# OpenReview forum: "Filters, Thresholds, and Geodesic Distances for Scribble-based Interactive Segmentation of Medical Images"
_thecvf.com/CVPR/2024/Workshop/MedSAMonLaptop — CVPR24 MedSAMonLaptop_

### Official Review · Reviewer_CLJC · 2024-06-14
**A good strategy that analyzes the characteristics of image modalities well**

**Rating:** 8
**Confidence:** 4

**Review:**

I was impressed with the way you analyzed and handled the characteristics of the image forms in the scribble-based interactive segmentation technique, respectively.
There appears to be an error in the reference to figures in section 4.8. Also, providing the sources of the various datasets used in the challenge would improve the clarity and reproducibility of the study.

---

### Official Review · Reviewer_PVre · 2024-06-16
**Traditional segmentation methods outperforming complex models in specific medical imaging modalities but lacking generalizability**

**Rating:** 7
**Confidence:** 3

**Review:**

The paper explores the efficacy of traditional segmentation methods such as thresholding, Meijering filters, and Geodesic Distance Transforms in comparison to the more recent MedSAM model, specifically for scribble-based interactive segmentation of medical images. Focusing on modalities like fundus, OCT, PET, and microscopy, the study highlights the potential advantages of simpler models over foundation models in terms of segmentation accuracy and computational efficiency. The paper's methodological approach, examining each imaging modality individually, provides a clear understanding of where traditional methods can outshine more complex models. However, the use of different preprocessing methods for each modality seems to contradict the overarching goal of developing a generalizable foundation model, potentially limiting its broad applicability.

---

### Decision · Program_Chairs · 2024-10-01

Accept